# Spatially Aggregated Gaussian Processes with Multivariate Areal Outputs

**Yusuke Tanaka[1,3], Toshiyuki Tanaka[3], Tomoharu Iwata[2], Takeshi Kurashima[1],**
**Maya Okawa[1], Yasunori Akagi[1], Hiroyuki Toda[1]**
[1]NTT Service Evolution Labs., [2]NTT Communication Science Labs., [3]Kyoto University
`{yusuke.tanaka.rh,tomoharu.iwata.gy,takeshi.kurashima.uf, maya.ookawa.af,`
`yasunori.akagi.cu,hiroyuki.toda.xb}@hco.ntt.co.jp, tt@i.kyoto-u.ac.jp`

## Abstract

We propose a probabilistic model for inferring the multivariate function from multiple areal data sets with various granularities. Here, the areal data are observed not at location points but at regions. Existing regression-based models can only utilize the sufficiently fine-grained auxiliary data sets on the same domain (e.g., a city). With the proposed model, the functions for respective areal data sets are assumed to be a multivariate dependent Gaussian process (GP) that is modeled as a linear mixing of independent latent GPs. Sharing of latent GPs across multiple areal data sets allows us to effectively estimate the spatial correlation for each areal data set; moreover it can easily be extended to transfer learning across multiple domains. To handle the multivariate areal data, we design an observation model with a spatial aggregation process for each areal data set, which is an integral of the mixed GP over the corresponding region. By deriving the posterior GP, we can predict the data value at any location point by considering the spatial correlations and the dependences between areal data sets, simultaneously. Our experiments on real-world data sets demonstrate that our model can 1) accurately refine coarse-grained areal data, and 2) offer performance improvements by using the areal data sets from multiple domains.

## 1 Introduction

Governments and other organizations are now collecting data from cities on items such as poverty rate, air pollution, crime, energy consumption, and traffic flow. These data play a crucial role in improving the life quality of citizens in many aspects including socio-economics [23, 24], public security [2, 32], public health [12], and urban planning [38]. For instance, the spatial distribution of poverty is helpful in identifying key regions that require intervention in a city; it makes it easier to optimize resource allocation for remedial action.

In practice, the data collected from cities are often spatially aggregated, e.g., averaged over a region; thus only *areal data* are available; observations are not associated with location points but with regions. Figure 1 shows an example of areal data, which is the distribution of poverty rate in New York City, where darker hues represent regions with higher rates. This poverty rate data set was actually obtained via household surveys taken over the whole city. The survey results are aggregated over predefined regions [24]. The problem addressed herein is to infer the *function* from the areal data; once we have the function we can predict data values at any location point. Solving this problem allows us to obtain spatially-specific information about cities; it is useful for finding key pin-point regions efficiently.

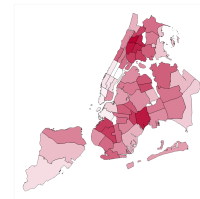

Figure 1: Areal data

One promising approach to address this problem is to utilize a wide variety of data sets from the same domain (e.g., a city). Existing regression-based models learn relationships between target data and auxiliary data sets [14, 18, 21, 24, 28]. These models, however, assume that the auxiliary data sets have sufficiently fine spatial granularity (e.g., 1 km × 1 km grid cells); unfortunately, many areal data sets are actually associated with large regions (e.g., zip code and police precinct). These models cannot then make full use of the coarse-grained auxiliary data sets. Another important drawback of all the prior works is that their performance in refining the areal data is suspect if we have only a few data sets available for the domain.

In this paper, we propose a probabilistic model, called *Spatially Aggregated Gaussian Processes (SAGP)* herein after, that can infer the multivariate function from multiple areal data sets with various granularities. In SAGP, the functions for the areal data sets are assumed to be a multivariate dependent Gaussian process (GP) that is modeled as a linear mixing of independent latent GPs. The latent GPs are shared among all areal data sets in the target domain, which is expected to effectively learn the spatial correlation for each data set even if the number of observations in a data set is small; that is, a data set associated with a coarse-grained region. Since the areal data are identified by regions, not by location points, we introduce an observation model with the spatial aggregation process, in which areal observations are assumed to be calculated by integrating the mixed GP over the corresponding region; then the covariance between regions is given by the double integral of the covariance function over the corresponding pair of regions. This allows us to accurately evaluate the covariance between regions from a consideration of region shape. Thus the proposal is very helpful if there are irregularly shaped regions (e.g., extremely elongated) in the input data.

The mechanism adopted in SAGP for sharing latent processes is also advantageous in that it makes it straightforward to utilize data sets from multiple domains. This allows our model to learn the spatial correlation for each data set by sharing the latent GPs among all areal data sets from multiple domains; SAGP remains applicable even if we have only a few data sets available for a single domain.

The inference of SAGP is based on a Bayesian inference procedure. The model parameters can be estimated by maximizing the marginal likelihood, in which all the GPs are analytically integrated out. By deriving the posterior GP, we can predict the data value at any location point considering the spatial correlations and the dependences between areal data sets, simultaneously.

The major contributions of this paper are as follows:

- We propose SAGP, a novel multivariate GP model that is defined by mixed latent GPs; it incorporates aggregation processes for handling multivariate areal data.

- We develop a parameter estimation procedure based on the marginal likelihood in which latent GPs are analytically integrated out. This is the first explicit derivation of the posterior GP given multivariate areal data; it allows the prediction of values at any location point.

- We conduct experiments on multiple real-world data sets from urban cities; the results show that our model can 1) accurately refine coarse-grained areal data, and 2) improve refinement performances by utilizing areal data sets from multiple cities.

## 2   Related Work

Related works can be roughly categorized into two approaches: 1) regression-based model and 2) multivariate model. The major difference between them is as follows: Denoting $\boldsymbol{y}^{\text{t}}$ and $\boldsymbol{y}$ as target data and auxiliary data, respectively, the aim of the first approach is to design a conditional distribution $p(\boldsymbol{y}^{\text{t}}|\boldsymbol{y})$; the second approach designs a joint distribution $p(\boldsymbol{y}^{\text{t}}, \boldsymbol{y})$.

**Regression-based models.** A related problem has been addressed in the spatial statistics community under the name of *downscaling*, *spatial disaggregation*, *areal interpolation*, or *fine-scale modeling* [8], and this has attracted great interest in many disciplines such as socio-economics [2, 24], agricultural economics [11, 36], epidemiology [27], meteorology [33, 35], and geographical information systems (GIS) [7]. The problem of predicting point-referenced data from areal observations is also related to the *change of support problem* in geostatistics [8]. Regression-based models have been developed for refining coarse-grained target data via the use of multiple auxiliary data sets that have fine granularity (e.g., 1 km × 1 km grid cells) [18, 21]. These models learn the regression coefficients for the auxiliary data sets under the spatial aggregation constraints that encourage consistency between fine- and coarse-grained target data. The aggregation constraints have been incorporated via *block kriging* [5]

or transformations of Gaussian process (GP) priors [19, 25]. There have been a number of advanced models that offer a fully Bayesian inference [13, 29, 34] or a variational inference [14] for model parameters. The task addressed in these works is to refine the coarse-grained target data on the assumption that the fine-grained auxiliary data are available; however, the areal data available on a city are actually associated with various geographical partitions (e.g., police precinct), thus one might not be able to obtain the fine-grained auxiliary data. In that case, these models cannot make full use of the auxiliary data sets with various granularities, which contain the coarse-grained auxiliary data.

A GP-based model was recently proposed for refining coarse-grained areal data by utilizing auxiliary data sets with various granularities [28]. In this model, GP regression is first applied to each auxiliary data set for deriving a predictive distribution defined on the continuous space; this conceptually corresponds to spatial interpolation. By hierarchically incorporating the predictive distributions into the model, the regression coefficients can be learned on the basis of not only the strength of relationship with the target data but also the level of spatial granularity. A major disadvantage of this model is that the spatial interpolation is separately conducted for each auxiliary data set, which makes it difficult to accurately interpolate the coarse-grained auxiliary data due to the data sparsity issue; this model fails to fully use the coarse-grained data.

In addition, these regression-based models (e.g., [14, 21, 28]) do not consider the spatial aggregation constraints for the auxiliary data sets. This is a critical issue in estimating the multivariate function from multiple areal data sets, the problem focused in this paper.

Different from the regression-based models, we design a joint distribution that incorporates the spatial aggregation process for all areal data sets (i.e., for both target and auxiliary data sets). The proposed model infers the multivariate function while considering the spatial aggregation constraints for respective areal data sets. This allows us to effectively utilize all areal data sets with various granularities for the data refinement even if some auxiliary data sets have coarse granularity.

**Multivariate models.** The proposed model builds closely upon recent studies in multivariate spatial modeling, which model the joint distribution of multiple outputs. Many geostatistics studies use the classical method of *co-kriging* for predicting multivariate spatial data [20]; this method is, however, problematic in that it is unclear how to define cross-covariance functions that determine the dependences between data sets [6]. In the machine learning community, there has been growing interest in multivariate GPs [22], in which dependences between data sets are introduced via methodologies such as process convolution [4, 10], latent factor modeling [16, 30], and multi-task learning [3, 17]. The linear model of coregionalization (LMC) is one of the most widely-used approaches for constructing a multivariate function; the outputs are expressed as linear combinations of independent latent functions [39]. The semiparametric latent factor model (SLFM) is an instance of LMC, in which latent functions are defined by GPs [30]. Unfortunately, these multivariate models cannot be straightforwardly used for modeling the areal data we focus on, because they assume that the data samples are observed at location points; namely they do not have an essential mechanism, i.e., the spatial aggregation constraints, for handling data that has been aggregated over regions.

The proposed model is an extension of SLFM. To handle the multivariate areal data, we newly introduce an observation model with the spatial aggregation process for all areal data sets; this is represented by the integral of the mixed GP over each corresponding region, as in block kriging. We also derive the posterior GP, which enables us to obtain the multivariate function from the observed areal data sets. Furthermore, the sharing of key information (i.e., covariance function) can be used for *transfer learning* across a wide variety of areal data sets; this allows our model to robustly estimate the spatial correlations for areal data sets and to support areal data sets from multiple domains.

Multi-task GP models have recently and independently been proposed for addressing similar problems [9, 37]. Main differences of our work from them are as follows: 1) Explicit derivation of the posterior GP given multivariate areal data; 2) transfer learning across multiple domains; 3) extensive experiments on real-world data sets defined on the two-dimensional input space.

# 3   Proposed Model

We propose SAGP (Spatially Aggregated Gaussian Processes), a probabilistic model for inferring the multivariate function from areal data sets with various granularities. We first consider a formulation in the case of a single domain, then we mention an extension to the case of multiple domains.

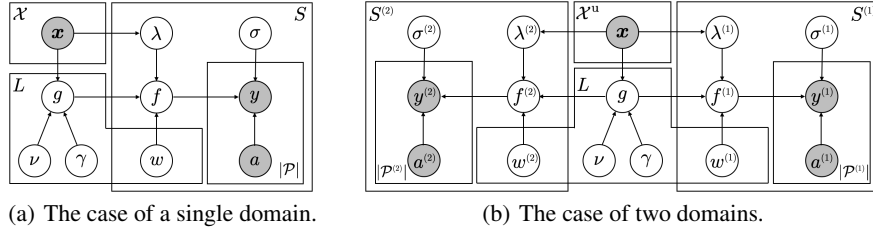

(a) The case of a single domain.  (b) The case of two domains.

Figure 2: Graphical model representation of SAGP.

**Areal data.** We start by describing the areal data this study focuses on. For simplicity, let us consider the case of a single domain (e.g., a city). Assume that we have a wide variety of areal data sets from the same domain and each data set is associated with one of the geographical partitions that have various granularities. Let $S$ be the number of kinds of areal data sets. Let $\mathcal{X} \subset \mathbb{R}^2$ denote an input space (e.g., a total region of a city), and $\boldsymbol{x} \in \mathcal{X}$ denote an input variable, represented by its coordinates (e.g., latitude and longitude). For $s = 1, \ldots, S$, the partition $\mathcal{P}_s$ of $\mathcal{X}$ is a collection of disjoint subsets, called *regions*, of $\mathcal{X}$. Let $|\mathcal{P}_s|$ be the number of regions in $\mathcal{P}_s$. For $n = 1, \ldots, |\mathcal{P}_s|$, let $\mathcal{R}_{s,n} \in \mathcal{P}_s$ denote the $n$-th region in $\mathcal{P}_s$. Each areal observation is represented by the pair $(\mathcal{R}_{s,n}, y_{s,n})$, where $y_{s,n} \in \mathbb{R}$ is a value associated with the $n$-th region $\mathcal{R}_{s,n}$. Suppose that we have $S$ areal data sets $\{(\mathcal{R}_{s,n}, y_{s,n}) \mid s = 1, \ldots, S; n = 1, \ldots, |\mathcal{P}_s|\}$.

**Formulation for the case of a single domain.** In the proposed model, the functions for the respective areal data sets on the continuous space are assumed to be the dependent Gaussian process (GP) with multivariate outputs. We first construct the multivariate dependent GP by linearly mixing some independent latent GPs. Consider $L$ independent GPs,

$$g_l(\boldsymbol{x}) \sim \mathcal{GP}\left(\nu_l(\boldsymbol{x}), \gamma_l(\boldsymbol{x}, \boldsymbol{x}')\right), \quad l = 1, \ldots, L, \tag{1}$$

where $\nu_l(\boldsymbol{x}) : \mathcal{X} \to \mathbb{R}$ and $\gamma_l(\boldsymbol{x}, \boldsymbol{x}') : \mathcal{X} \times \mathcal{X} \to \mathbb{R}$ are a mean function and a covariance function, respectively, for the $l$-th latent GP $g_l(\boldsymbol{x})$, both of which are assumed integrable. Defining $f_s(\boldsymbol{x})$ as the $s$-th GP, the $S$-dimensional dependent GP $\boldsymbol{f}(\boldsymbol{x}) = (f_1(\boldsymbol{x}), \ldots, f_S(\boldsymbol{x}))^\top$ is assumed to be modeled as a linear mixing of the $L$ independent latent GPs, then $\boldsymbol{f}(\boldsymbol{x})$ is given by

$$\boldsymbol{f}(\boldsymbol{x}) = \mathbf{W}\boldsymbol{g}(\boldsymbol{x}) + \boldsymbol{n}(\boldsymbol{x}), \tag{2}$$

where $\boldsymbol{g}(\boldsymbol{x}) = (g_1(\boldsymbol{x}), \ldots, g_L(\boldsymbol{x}))^\top$, $\mathbf{W}$ is an $S \times L$ weight matrix whose $(s, l)$-entry $w_{s,l} \in \mathbb{R}$ is the weight of the $l$-th latent GP in the $s$-th data set, and $\boldsymbol{n}(\boldsymbol{x}) \sim \mathcal{GP}(\mathbf{0}, \boldsymbol{\Lambda}(\boldsymbol{x}, \boldsymbol{x}'))$ is an $S$-dimensional zero-mean Gaussian noise process. Here, $\mathbf{0}$ is a column vector of 0's and $\boldsymbol{\Lambda}(\boldsymbol{x}, \boldsymbol{x}') = \mathrm{diag}(\lambda_1(\boldsymbol{x}, \boldsymbol{x}'), \ldots, \lambda_S(\boldsymbol{x}, \boldsymbol{x}'))$ with $\lambda_s(\boldsymbol{x}, \boldsymbol{x}') : \mathcal{X} \times \mathcal{X} \to \mathbb{R}$ being a covariance function for the $s$-th Gaussian noise process. By integrating out $\boldsymbol{g}(\boldsymbol{x})$, the multivariate GP $\boldsymbol{f}(\boldsymbol{x})$ is given by

$$\boldsymbol{f}(\boldsymbol{x}) \sim \mathcal{GP}\left(\boldsymbol{m}(\boldsymbol{x}), \mathbf{K}(\boldsymbol{x}, \boldsymbol{x}')\right), \tag{3}$$

where the mean function $\boldsymbol{m}(\boldsymbol{x}) : \mathcal{X} \to \mathbb{R}^S$ is given by $\boldsymbol{m}(\boldsymbol{x}) = \mathbf{W}\boldsymbol{\nu}(\boldsymbol{x})$. The covariance function $\mathbf{K}(\boldsymbol{x}, \boldsymbol{x}') : \mathcal{X} \times \mathcal{X} \to \mathbb{R}^{S \times S}$ is given by $\mathbf{K}(\boldsymbol{x}, \boldsymbol{x}') = \mathbf{W}\boldsymbol{\Gamma}(\boldsymbol{x}, \boldsymbol{x}')\mathbf{W}^\top + \boldsymbol{\Lambda}(\boldsymbol{x}, \boldsymbol{x}')$. Here, $\boldsymbol{\nu}(\boldsymbol{x}) = (\nu_1(\boldsymbol{x}), \ldots, \nu_L(\boldsymbol{x}))^\top$ and $\boldsymbol{\Gamma}(\boldsymbol{x}, \boldsymbol{x}') = \mathrm{diag}(\gamma_1(\boldsymbol{x}, \boldsymbol{x}'), \ldots, \gamma_L(\boldsymbol{x}, \boldsymbol{x}'))$. The derivation of (3) is described in Appendix A of Supplementary Material. The $(s, s')$-entry of $\mathbf{K}(\boldsymbol{x}, \boldsymbol{x}')$ is given by

$$k_{s,s'}(\boldsymbol{x}, \boldsymbol{x}') = \delta_{s,s'}\lambda_s(\boldsymbol{x}, \boldsymbol{x}') + \sum_{l=1}^{L} w_{s,l}w_{s',l}\gamma_l(\boldsymbol{x}, \boldsymbol{x}'), \tag{4}$$

where $\delta_{\bullet,\bullet}$ represents Kronecker's delta; $\delta_{Z,Z'} = 1$ if $Z = Z'$ and $\delta_{Z,Z'} = 0$ otherwise. The covariance function (4) for the multivariate GP $\boldsymbol{f}(\boldsymbol{x})$ is represented by the linear combination of the covariance functions $\{\gamma_l(\boldsymbol{x}, \boldsymbol{x}')\}_{l=1}^{L}$ for the latent GPs. The covariance functions for latent GPs are shared among all areal data sets, which allows us to effectively learn the spatial correlation for each data set by considering the dependences between data sets; this is advantageous in the case where the number of observations is small, that is, the spatial granularity of the areal data is coarse. In this paper we focus on the case $L < S$, with the aim of reducing the number of free parameters as this helps to avoid overfitting [30].

The areal data are not associated with location points but with regions, and their observations are obtained by spatially aggregating the original data. To handle the multivariate areal data, we

design an observation model with a spatial aggregation process for each of the areal data sets. Let $\boldsymbol{y}_s = (y_{s,1}, \ldots, y_{s,|\mathcal{P}_s|})$ be a $|\mathcal{P}_s|$-dimensional vector consisting of the areal observations for the $s$-th areal data set. Let $\boldsymbol{y} = (\boldsymbol{y}_1, \boldsymbol{y}_2, \ldots, \boldsymbol{y}_S)^\top$ denote an $N$-dimensional vector consisting of the observations for all areal data sets, where $N = \sum_{s=1}^{S} |\mathcal{P}_s|$ is the total number of areal observations. Each areal observation is assumed to be obtained by integrating the mixed GP $\boldsymbol{f}(\boldsymbol{x})$ over the corresponding region; $\boldsymbol{y}$ is generated from the Gaussian distribution[1],

$$\boldsymbol{y} \mid \boldsymbol{f}(\boldsymbol{x}) \sim \mathcal{N}\left(\boldsymbol{y} \;\Big|\; \int_{\mathcal{X}} \mathbf{A}(\boldsymbol{x})\boldsymbol{f}(\boldsymbol{x})\,d\boldsymbol{x}, \boldsymbol{\Sigma}\right), \tag{5}$$

where $\mathbf{A}(\boldsymbol{x}) : \mathcal{X} \to \mathbb{R}^{N \times S}$ is represented by

$$\mathbf{A}(\boldsymbol{x}) = \begin{pmatrix} \boldsymbol{a}_1(\boldsymbol{x}) & \mathbf{0} & \cdots & \mathbf{0} \\ \mathbf{0} & \boldsymbol{a}_2(\boldsymbol{x}) & \cdots & \mathbf{0} \\ \vdots & \vdots & \ddots & \vdots \\ \mathbf{0} & \mathbf{0} & \cdots & \boldsymbol{a}_S(\boldsymbol{x}) \end{pmatrix}, \tag{6}$$

in which $\boldsymbol{a}_s(\boldsymbol{x}) = \left(a_{s,1}(\boldsymbol{x}), \ldots, a_{s,|\mathcal{P}_s|}(\boldsymbol{x})\right)^\top$, whose entry $a_{s,n}(\boldsymbol{x})$ is a nonnegative weight function for spatial aggregation over region $\mathcal{R}_{s,n}$. This formulation does not depend on the particular choice of $\{a_{s,n}(\boldsymbol{x})\}$, provided that they are integrable. If one takes, for region $\mathcal{R}_{s,n}$,

$$a_{s,n}(\boldsymbol{x}) = \frac{\mathbb{1}(\boldsymbol{x} \in \mathcal{R}_{s,n})}{\int_{\mathcal{X}} \mathbb{1}(\boldsymbol{x}' \in \mathcal{R}_{s,n})\,d\boldsymbol{x}'}, \tag{7}$$

where $\mathbb{1}(\bullet)$ is the indicator function; $\mathbb{1}(Z) = 1$ if $Z$ is true and $\mathbb{1}(Z) = 0$ otherwise, then $y_{s,n}$ is the average of $f_s(\boldsymbol{x})$ over $\mathcal{R}_{s,n}$. We may also consider other aggregation processes to suit the property of the areal observations, including simple summation and population-weighted averaging over $\mathcal{R}_{s,n}$. $\boldsymbol{\Sigma} = \mathrm{diag}(\sigma_1^2 \mathbf{I}, \ldots, \sigma_S^2 \mathbf{I})$ in (5) is an $N \times N$ block diagonal matrix, where $\sigma_s^2$ is the noise variance for the $s$-th GP, and $\mathbf{I}$ is the identity matrix. Figure 2(a) shows a graphical model representation of SAGP, where shaded and unshaded nodes indicate observed and latent variables, respectively.

**Extension to the case of multiple domains.** It is possible to apply SAGP to areal data sets from multiple domains by assuming that observations are conditionally independent given the latent GPs $\{g_l(\boldsymbol{x})\}_{l=1}^{L}$. The graphical model representation of SAGP shown in Figure 2(b) is for the case of two domains. The superscript in Figure 2(b) is the domain index, and $\mathcal{X}^{\mathrm{u}}$ is the union of the input spaces for both domains. Although $\boldsymbol{y}^{(1)}$ and $\boldsymbol{y}^{(2)}$ in Figure 2(b) are not directly correlated across domains, the shared covariance functions $\{\gamma_l(\boldsymbol{x}, \boldsymbol{x}')\}_{l=1}^{L}$ for the latent GPs can be learned by transfer learning based on the data sets from multiple domains; thus the spatial correlation for each data set could be more appropriately output via the covariance functions, even if we have only a few data sets available for a single domain. SAGP can be extended to the case of more domains in a similar fashion.

## 4 Inference

Given the areal data sets, we aim to derive the posterior GP on the basis of a Bayesian inference procedure. The posterior GP can be used for predicting data values at any location point in the continuous space. The model parameters, $\mathbf{W}$, $\boldsymbol{\Lambda}(\boldsymbol{x}, \boldsymbol{x}')$, $\boldsymbol{\Sigma}$, $\boldsymbol{\nu}(\boldsymbol{x})$, $\boldsymbol{\Gamma}(\boldsymbol{x}, \boldsymbol{x}')$, are estimated by maximizing the marginal likelihood, in which multivariate GP $\boldsymbol{f}(\boldsymbol{x})$ is analytically integrated out; we then construct the posterior GP by using the estimated parameters.

**Marginal likelihood.** Consider the case of a single domain. Given the areal data $\boldsymbol{y}$, the marginal likelihood is given by

$$p(\boldsymbol{y}) = \mathcal{N}\left(\boldsymbol{y} \mid \boldsymbol{\mu}, \mathbf{C}\right), \tag{8}$$

where we analytically integrate out the GP prior $\boldsymbol{f}(\boldsymbol{x})$. Here, $\boldsymbol{\mu}$ is an $N$-dimensional mean vector represented by

$$\boldsymbol{\mu} = \int_{\mathcal{X}} \mathbf{A}(\boldsymbol{x})\boldsymbol{m}(\boldsymbol{x})\,d\boldsymbol{x}, \tag{9}$$

which is the integral of the mean function $m(x)$ over the respective regions for all areal data sets. $\mathbf{C}$ is an $N \times N$ covariance matrix represented by

$$\mathbf{C} = \iint_{\mathcal{X} \times \mathcal{X}} \mathbf{A}(x) \mathbf{K}(x, x') \mathbf{A}(x')^{\top} \, dx \, dx' + \boldsymbol{\Sigma}. \tag{10}$$

It is an $S \times S$ block matrix whose $(s, s')$-th block $\mathbf{C}_{s,s'}$ is a $|\mathcal{P}_s| \times |\mathcal{P}_{s'}|$ matrix represented by

$$\mathbf{C}_{s,s'} = \iint_{\mathcal{X} \times \mathcal{X}} k_{s,s'}(x, x') a_s(x) a_{s'}(x')^{\top} \, dx \, dx' + \delta_{s,s'} \sigma_s^2 \mathbf{I}. \tag{11}$$

Equation (11) provides the region-to-region covariance matrices in the form of the double integral of the covariance function $k_{s,s'}(x, x')$ over the respective pairs of regions in $\mathcal{P}_s \times \mathcal{P}_{s'}$; this conceptually corresponds to aggregation of the covariance function values that are calculated at the infinite pairs of location points in the corresponding regions. Since the integrals over regions cannot be calculated analytically, in practice we use a numerical approximation of these integrals. Details are provided at the end of this section. This formulation allows for accurately evaluating the covariance between regions considering their shapes; this is extremely helpful as some input data are likely to originate from irregularly shaped regions (e.g., extremely elongated). By maximizing the logarithm of the marginal likelihood (8), we can estimate the parameters of SAGP.

**Transfer learning across multiple domains.** Consider the case of $V$ domains. Let $\{y^{(v)}\}_{v=1}^V$ denote the collection of data sets for the $V$ domains. In SAGP, the observations for different domains are assumed to be conditionally independent given the shared latent GPs $\{g_l(x)\}_{l=1}^L$; the marginal likelihood for $V$ domains is thus given by the product of those for the $V$ domains:

$$p\left(y^{(1)}, y^{(2)}, \ldots, y^{(V)}\right) = \prod_{v=1}^V \mathcal{N}\left(y^{(v)} \mid \boldsymbol{\mu}^{(v)}, \mathbf{C}^{(v)}\right), \tag{12}$$

where $\boldsymbol{\mu}^{(v)}$ and $\mathbf{C}^{(v)}$ are the mean vector and the covariance matrix for the $v$-th domain, respectively. Estimation of model parameters based on (12) allows for transfer learning across the areal data sets from multiple domains via the shared covariance functions.

**Posterior GP.** We have only to consider the case of a single domain, because the derivation of the posterior GP can be conducted independently for each domain. Given the areal data $y$ and the estimated parameters, the posterior GP $f^*(x)$ is given by

$$f^*(x) \sim \mathcal{GP}\left(m^*(x), \mathbf{K}^*(x, x')\right), \tag{13}$$

where $m^*(x) : \mathcal{X} \to \mathbb{R}^S$ and $\mathbf{K}^*(x, x') : \mathcal{X} \times \mathcal{X} \to \mathbb{R}^{S \times S}$ are the mean function and the covariance function for $f^*(x)$, respectively. Defining $\mathbf{H}(x) : \mathcal{X} \to \mathbb{R}^{N \times S}$ as

$$\mathbf{H}(x) = \int_{\mathcal{X}} \mathbf{A}(x') \mathbf{K}(x', x) \, dx', \tag{14}$$

which consists of the point-to-region covariances, which are the covariances between any location point $x$ and the respective regions in all areal data sets, the mean function $m^*(x)$ and the covariance function $\mathbf{K}^*(x, x')$ are given by

$$m^*(x) = m(x) + \mathbf{H}(x)^{\top} \mathbf{C}^{-1}(y - \boldsymbol{\mu}), \tag{15}$$

$$\mathbf{K}^*(x, x') = \mathbf{K}(x, x') - \mathbf{H}(x)^{\top} \mathbf{C}^{-1} \mathbf{H}(x'), \tag{16}$$

respectively. We can predict the data value at any location point by using the mean function (15). The second term in (15) shows that the predictions are calculated by considering the spatial correlations and the dependences between areal data sets, simultaneously. By using the covariance function (16), we can also evaluate the prediction uncertainty. Derivation of the posterior GP is detailed in Appendix B of Supplementary Material.

**Approximation of the integral over regions.** The integrals over regions in (9), (11), and (14) cannot be performed analytically; thus we approximate these integrals by using sufficiently fine-grained square grid cells. We divide input space $\mathcal{X}$ into square grid cells, and take $\mathcal{G}_{s,n}$ to be the set of grid points that are contained in region $\mathcal{R}_{s,n}$. Let us consider the approximation of the integral in the covariance matrix (11). The $(n, n')$-entry $C_{s,s'}(n, n')$ of $\mathbf{C}_{s,s'}$ is approximated as follows:

$$\mathbf{C}_{s,s'}(n,n') = \iint_{\mathcal{X}\times\mathcal{X}} k_{s,s'}(\boldsymbol{x},\boldsymbol{x}')a_{s,n}(\boldsymbol{x})a_{s',n'}(\boldsymbol{x}')\,d\boldsymbol{x}\,d\boldsymbol{x}' + \delta_{s,s'}\delta_{n,n'}\sigma_s \qquad (17)$$

$$\approx \frac{1}{|\mathcal{G}_{s,n}|}\frac{1}{|\mathcal{G}_{s',n'}|}\sum_{i\in\mathcal{G}_{s,n}}\sum_{j\in\mathcal{G}_{s',n'}} k_{s,s'}(i,j) + \delta_{s,s'}\delta_{n,n'}\sigma_s, \qquad (18)$$

where we use the formulation of the region-average-observation model (7). The integrals in (9) and (14) can be approximated in a similar way. Letting $|\mathcal{G}|$ denote the number of all grid points, the computational complexity of $\mathbf{C}_{s,s'}$ (11) is $O(|\mathcal{G}|^2)$; assuming the constant weight $a_{s,n}(\boldsymbol{x}) = a_{s,n}$ (e.g., region average), the computational complexity can be reduced to $O(|\mathcal{P}_s||\mathcal{P}_{s'}||\mathcal{D}|)$, where $|\mathcal{D}|$ is the cardinality of the set of distinct distance values between grid points. Here, we use the property that $k_{s,s'}(i,j)$ in (18) depends only on the distance between $i$ and $j$. This is useful for reducing the computation time and the memory requirement. The average computation times for inference were 1728.2 and 115.1 seconds for the data sets from New York City and Chicago, respectively; the experiments were conducted on a 3.1 GHz Intel Core i7.

## 5 Experiments

**Data.** We evaluated SAGP using 10 and 3 real-world areal data sets from two cities, New York City and Chicago, respectively. They were obtained from NYC Open Data [2] and Chicago Data Portal [3]. We used a variety of areal data sets including poverty rate, air pollution rate, and crime rate. Each data set is associated with one of the predefined geographical partitions with various granularities: UHF42 (42), community district (59), police precinct (77), and zip code (186) in New York City; police precinct (25) and community district (77) in Chicago, where each number in parentheses denotes the number of regions in the corresponding partition. In the experiments, the data were normalized so that each variable in each city has zero mean and unit variance. Details about the real-world data sets are provided in Appendix D of Supplementary Material.

**Refinement task.** We examined the task of refining coarse-grained areal data by using multiple areal data sets with various granularities. To evaluate the performance in predicting the fine-grained areal data, we first picked up one target data set and used its coarser version for learning model parameters; then we predicted the original fine-grained target data by using the learned model. Note that the fine-grained target data was used only for evaluating the refinement performance; we did not use them in the inference process. The target data sets were poverty rate (5, 59), PM2.5 (5, 42), crime (5, 77) in New York City and poverty rate (9, 77) in Chicago, where each pair of numbers in parentheses denotes the numbers of regions in the coarse- and the fine-grained partitions, respectively. Defining $s'$ as the index of the target data set, the evaluation metric is the mean absolute percentage error (MAPE) of the fine-grained target values, $\frac{1}{|\mathcal{P}_{s'}|}\sum_{n\in\mathcal{P}_{s'}}\left|(y_{s',n}^{\text{true}} - y_{s',n}^*)/y_{s',n}^{\text{true}}\right|$, where $y_{s',n}^{\text{true}}$ is the true value associated with the $n$-th region in the target fine-grained partition; $y_{s',n}^*$ is its predicted value, obtained by integrating the $s'$-th function $f_{s'}^*(\boldsymbol{x})$ of the posterior GP $\boldsymbol{f}^*(\boldsymbol{x})$ (13) over the corresponding target fine-grained region.

**Setup of the proposed model.** In our experiments, we used zero-mean Gaussian processes as the latent GPs $\{g_l(\boldsymbol{x})\}_{l=1}^L$, i.e., $\nu_l(\boldsymbol{x}) = 0$ for $l = 1, \ldots, L$. We used the following squared-exponential kernel as the covariance function for the latent GPs, $\gamma_l(\boldsymbol{x},\boldsymbol{x}') = \alpha_l^2 \exp\left(-\|\boldsymbol{x}-\boldsymbol{x}'\|^2/2\beta_l^2\right)$, where $\alpha_l^2$ is a signal variance that controls the magnitude of the covariance, $\beta_l$ is a scale parameter that determines the degrees of spatial correlation, and $\|\cdot\|$ is the Euclidean norm. Here, we set $\alpha_l^2 = 1$ because the variance can already be represented by scaling the columns of $\mathbf{W}$. For simplicity, the covariance function for the Gaussian noise process $\boldsymbol{n}(\boldsymbol{x},\boldsymbol{x}')$ is set to $\boldsymbol{\Lambda}(\boldsymbol{x},\boldsymbol{x}') = \text{diag}(\lambda_1^2\delta(\boldsymbol{x}-\boldsymbol{x}'),\ldots,\lambda_S^2\delta(\boldsymbol{x}-\boldsymbol{x}'))$, where $\delta(\bullet)$ is Dirac's delta function. The model parameters, $\mathbf{W}, \{\lambda_s\}_{s=1}^S$, $\boldsymbol{\Sigma}, \{\beta_l\}_{l=1}^L$, were learned by maximizing the logarithm of the marginal likelihood (8) or (12) using the L-BFGS method [15] implemented in SciPy (https://www.scipy.org/). For approximating the integral over regions (see (18)), we divided a total region of each city into sufficiently fine-grained square grid cells, the size of which was 300 m × 300 m for both cities; the resulting sets of grid points $\mathcal{G}$ for New York City and Chicago consisted of 9,352 and 7,400 grid points, respectively. The number $L$ of the latent GPs was chosen from $\{1,\ldots,S\}$ via leave-one-out cross-validation [1]; the

[2]https://opendata.cityofnewyork.us
[3]https://data.cityofchicago.org/

Table 1: MAPE and standard errors for the prediction of fine-grained areal data (a single city). The numbers in parentheses denote the number $L$ of the latent GPs determined by the validation procedure. The single star ($\star$) and the double star ($\star\star$) indicate significant difference between SAGP and other models at the levels of $P$ values of $< 0.05$ and $< 0.01$, respectively.

| | New York City | | | Chicago |
| | Poverty rate | PM2.5 | Crime | Poverty rate |
| --- | --- | --- | --- | --- |
| GPR | $0.344 \pm 0.046$ (–) | $0.072 \pm 0.010$ (–) | $0.860 \pm 0.102$ (–) | $0.599 \pm 0.099$ (–) |
| 2-stage GP | $0.210 \pm 0.022$ (–) | $0.042 \pm 0.005$ (–) | $0.454 \pm 0.075$ (–) | $0.380 \pm 0.060$ (–) |
| SLFM | $0.207 \pm 0.025$ (4) | $0.036 \pm 0.005$ (6) | $0.401 \pm 0.053$ (2) | $0.335 \pm 0.052$ (2) |
| SAGP | $\mathbf{0.177 \pm 0.019^{\star\star}}$ (3) | $\mathbf{0.030 \pm 0.005^{\star}}$ (5) | $\mathbf{0.379 \pm 0.055^{\star\star}}$ (3) | $\mathbf{0.278 \pm 0.032^{\star\star}}$ (2) |

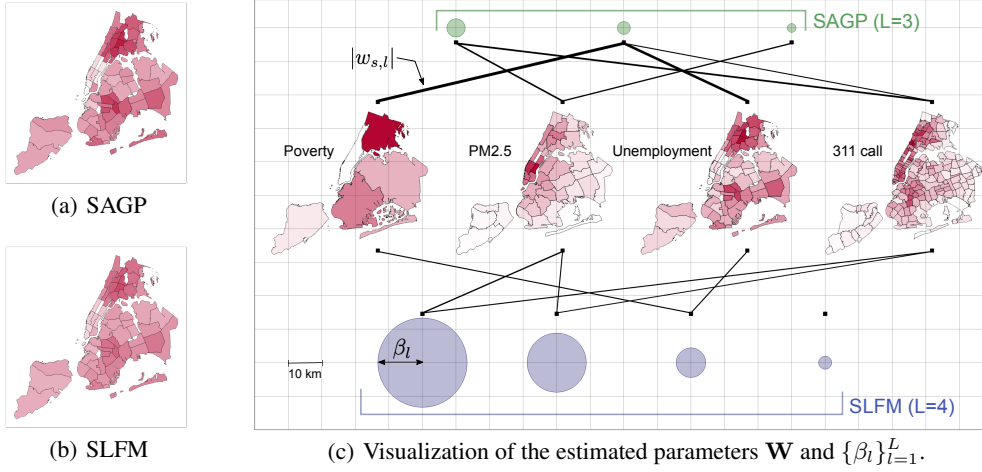

(a) SAGP

(b) SLFM

(c) Visualization of the estimated parameters $\mathbf{W}$ and $\{\beta_l\}_{l=1}^{L}$.

Figure 3: (a,b) Refined poverty rate data in NYC, and (c) Visualization of the estimated parameters when predicting the poverty rate data in NYC. The radii of green and blue circles equal the values of $\beta_l$ estimated by SAGP and SLFM, respectively. The edge widths are proportional to the absolute weights $|w_{s,l}|$ estimated by the respective models. Here, we omitted those edges whose absolute weights were lower than a threshold.

validation error was obtained using each held-out coarse-grained data value. Here, the validation was conducted on the basis of the coarse-grained target areal data; namely we did not use the fine-grained target data in the validation process.

**Baselines.** We compared the proposed model, SAGP, with naive Gaussian process regression (GPR) [22], two-stage GP-based model (2-stage GP) [28], and semiparametric latent factor model (SLFM) [30]. GPR predicts the fine-grained target data simply from just the coarse-grained target data. 2-stage GP is one of the latest regression-based models. SLFM is the multivariate GP model; SAGP is regarded as the extension of SLFM. GPR and SLFM assume that data samples are observed at location points. We thus associate each areal observation with the centroid of the region. This simplification is also used for modeling the auxiliary data sets in [28].

**Results for the case of a single city.** Table 1 shows MAPE and standard errors for GPR, 2-stage GP, SLFM, and SAGP. For all data sets, SAGP achieved better performance in refining coarse-grained areal data; the differences between SAGP and the baselines were statistically significant (Student's t-test). These results show that SAGP can utilize the areal data sets with various granularities from the same city to accurately predict the refined data. The results for all data sets from both cities are shown in Appendix E of Supplementary Material.

Figures 3(a) and 3(b) show the refinement results of SAGP and SLFM for the poverty rate data in New York City. Here, the predictive values of each model were normalized to the range $[0, 1]$, and darker hues represent regions with higher values. Compared with the true data in Figure 1, SAGP yielded more accurate fine-grained data than SLFM. Figure 3(c) visualizes the mixing weights $\mathbf{W}$ and the scale parameters $\{\beta_l\}_{l=1}^{L}$ estimated by SAGP and SLFM when predicting the fine-grained poverty rate data in New York City, where we picked up 4 areal data sets: Poverty rate, PM2.5, unemployment rate, and the number of 311 calls; their observations were also shown. One observes that the scale parameters estimated by SAGP are relatively small compared with those estimated

Table 2: MAPE and standard errors for the prediction of the fine-grained data (two cities).

| | Chicago |
|---|---|
| | Poverty rate |
| SLFM (trans) | $0.328 \pm 0.050$ (6) |
| SAGP (trans) | $\mathbf{0.219 \pm 0.023}$ (4) |

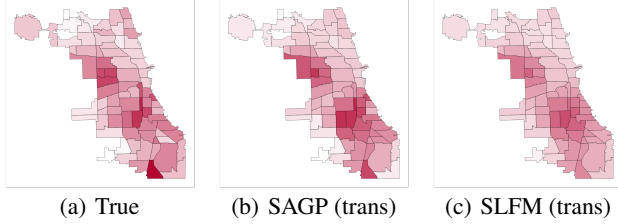

(a) True      (b) SAGP (trans)      (c) SLFM (trans)

Figure 5: Refined poverty rate data in Chicago.

by SLFM, presumably because the spatial aggregation process incorporated in SAGP effectively separates intrinsic spatial correlations and apparent smoothing effects due to the spatial aggregation to yield areal observations. A comparison of the estimated weights in Figure 3(c) shows that SAGP emphasized the useful dependences between data sets, e.g., the strong correlation between the poverty rate data and the unemployment rate data.

One benefit of SAGP is that all predictions associated with the target regions have uncertainty estimates, where the prediction variance can be calculated by integrating the covariance function $\mathbf{K}^*(\boldsymbol{x}, \boldsymbol{x}')$ (16) of the posterior GP (13) over the corresponding target region. Figure 4 visualizes the variance with SAGP in the prediction of the poverty rate in New York City and Chicago, respectively. One observes that the variances at the regions located at the edge of the city tend to have larger values compared with those inside the city. This is reasonable because extrapolation is gener-

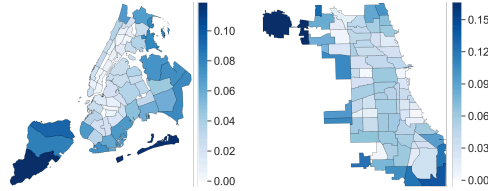

Figure 4: Variance of the posterior GP with SAGP for predicting the poverty rate in New York City (Left) and Chicago (Right), respectively.

ally more difficult than interpolation. These uncertainty estimates are useful in that the predictions may help guide policy and planning in a city even if validation of them is difficult.

**Results for the case of two cities.** SLFM and SAGP can be used for transfer learning across multiple cities, which is more advantageous in such a situation that we have only a few data sets available on a single city. We here show the results of refining the poverty rate data in Chicago with simultaneously utilizing the data sets from New York City. Table 2 shows MAPE and standard errors for SLFM (trans) and SAGP (trans). Comparing Tables 1 and 2, one observes that SAGP (trans) attained improved refinement performance compared with SLFM (trans) and models trained with only the data in a single city (i.e., Chicago). the differences between SAGP (trans) and the other models were statistically significant (Student's t-test, $P$ value of $< 0.01$). This result shows that SAGP (trans) transferred knowledge across the cities, and yielded better refinement results even if there are only a few data sets available on the target city. Figure 5 shows the refinement results for the poverty rate data in Chicago. We illustrate the true data on the left in Figure 5, and the predictions attained by SAGP (trans) and SLFM (trans) on the right. As shown, SAGP (trans) better identified the key regions compared with SLFM (trans).

## 6 Conclusion

This paper has proposed the Spatially Aggregated Gaussian Processes for inferring the multivariate function from multiple areal data sets with various granularities. To handle multivariate areal data, we design an observation model with the spatial aggregation process for each areal data set, which is the integral of the Gaussian process over the corresponding region. We have confirmed that our model can accurately refine the coarse-grained areal data, and improve the refinement performance by using the areal data sets from multiple cities.

There are several avenues that can be explored in future work. First, we can introduce nonlinear link functions, as in warped GP [26], and/or alternative likelihoods; this might help handle some kinds of observations (e.g., rates). Second, we can use scalable variational inference with inducing points, similar to [31], for large-scale data sets. Finally, our formulation provides a general framework for modeling aggregated data and offers a potential research direction; for instance, it has the ability to consider data aggregated on a higher dimensional input space, e.g., spatio-temporal aggregated data.

## Footnotes

[1]We here assume that the integral appearing in (5) is well-defined. It should be noted that without additional assumptions sample paths of a Gaussian process are in general not integrable. See Appendix C of Supplementary Material for discussion on the conditions under which the integral is well-defined.

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
