[Supplementary Material]

# Supplementary Material: Spatially Aggregated Gaussian Processes with Multivariate Areal Outputs

## A  Derivation of the multivariate GP $\boldsymbol{f}(\boldsymbol{x})$

In this appendix, we show that the process $\boldsymbol{f}(\boldsymbol{x})$ defined via (2) is itself a multivariate GP with mean function $\boldsymbol{m}(\boldsymbol{x}) = \mathbf{W}\boldsymbol{\nu}(\boldsymbol{x})$ and covariance function $\mathbf{K}(\boldsymbol{x}, \boldsymbol{x}') = \mathbf{W}\boldsymbol{\Gamma}(\boldsymbol{x}, \boldsymbol{x}')\mathbf{W}^\top + \boldsymbol{\Lambda}(\boldsymbol{x}, \boldsymbol{x}')$. To prove that $\boldsymbol{f}(\boldsymbol{x})$ is indeed a multivariate GP, one has only to show that, for an arbitrary $k \in \{1, 2, \ldots\}$ and an arbitrary set of $k$ points $\boldsymbol{x}_1, \ldots, \boldsymbol{x}_k \in \mathcal{X}$, $\bar{\boldsymbol{f}} = (\boldsymbol{f}(\boldsymbol{x}_1), \ldots, \boldsymbol{f}(\boldsymbol{x}_k))^\top$ is a multivariate Gaussian random variable. By the definition (2) of $\boldsymbol{f}(\boldsymbol{x})$, one has

$$\bar{\boldsymbol{f}} = (\mathbf{I} \otimes \mathbf{W})\bar{\boldsymbol{g}} + \bar{\boldsymbol{n}}, \tag{19}$$

where we let $\bar{\boldsymbol{g}} = (\boldsymbol{g}(\boldsymbol{x}_1), \ldots, \boldsymbol{g}(\boldsymbol{x}_k))^\top$ and $\bar{\boldsymbol{n}} = (\boldsymbol{n}(\boldsymbol{x}_1), \ldots, \boldsymbol{n}(\boldsymbol{x}_k))^\top$, and where $\otimes$ denotes the Kronecker product. By the definition of Gaussian processes, since $\boldsymbol{g}(\boldsymbol{x})$ and $\boldsymbol{n}(\boldsymbol{x})$ are Gaussian processes, $\bar{\boldsymbol{g}}$ and $\bar{\boldsymbol{n}}$ are multivariate Gaussian random variables. Since (19) shows that $\bar{\boldsymbol{f}}$ is a linear combination of the multivariate Gaussian random variables $\bar{\boldsymbol{g}}$ and $\bar{\boldsymbol{n}}$, it is itself multivariate Gaussian, irrespective of the choice of $\boldsymbol{x}_1, \ldots, \boldsymbol{x}_k$. This in turn shows that $\boldsymbol{f}(\boldsymbol{x})$ is again a multivariate Gaussian process.

Mean of $\boldsymbol{f}(\boldsymbol{x}_i)$ is given by

$$\mathbb{E}(\boldsymbol{f}(\boldsymbol{x}_i)) = \mathbf{W}\mathbb{E}(\boldsymbol{g}(\boldsymbol{x}_i)) = \mathbf{W}\boldsymbol{\nu}(\boldsymbol{x}_i). \tag{20}$$

Covariance of $\boldsymbol{f}(\boldsymbol{x}_i)$ and $\boldsymbol{f}(\boldsymbol{x}_j)$ is given by

$$\begin{aligned}
\text{Cov}(\boldsymbol{f}(\boldsymbol{x}_i), \boldsymbol{f}(\boldsymbol{x}_j)) &= \mathbb{E}((\boldsymbol{f}(\boldsymbol{x}_i) - \mathbf{W}\boldsymbol{\nu}(\boldsymbol{x}_i))(\boldsymbol{f}(\boldsymbol{x}_j) - \mathbf{W}\boldsymbol{\nu}(\boldsymbol{x}_j))^\top) \\
&= \mathbb{E}((\mathbf{W}(\boldsymbol{g}(\boldsymbol{x}_i) - \boldsymbol{\nu}(\boldsymbol{x}_i)) + \boldsymbol{n}(\boldsymbol{x}_i))(\mathbf{W}(\boldsymbol{g}(\boldsymbol{x}_j) - \boldsymbol{\nu}(\boldsymbol{x}_j)) + \boldsymbol{n}(\boldsymbol{x}_j))^\top) \\
&= \mathbf{W}\boldsymbol{\Gamma}(\boldsymbol{x}_i, \boldsymbol{x}_j)\mathbf{W}^\top + \boldsymbol{\Lambda}(\boldsymbol{x}_i, \boldsymbol{x}_j).
\end{aligned} \tag{21}$$

These show that the mean function $\boldsymbol{m}(\boldsymbol{x})$ and the covariance function $\mathbf{K}(\boldsymbol{x}, \boldsymbol{x}')$ of the multivariate Gaussian process $\boldsymbol{f}(\boldsymbol{x})$ are given by $\boldsymbol{m}(\boldsymbol{x}) = \mathbf{W}\boldsymbol{\nu}(\boldsymbol{x})$ and $\mathbf{K}(\boldsymbol{x}, \boldsymbol{x}') = \mathbf{W}\boldsymbol{\Gamma}(\boldsymbol{x}, \boldsymbol{x}')\mathbf{W}^\top + \boldsymbol{\Lambda}(\boldsymbol{x}, \boldsymbol{x}')$, respectively.

## B  Derivation of the posterior GP $\boldsymbol{f}^*(\boldsymbol{x})$

In this appendix, we derive the posterior Gaussian process $\boldsymbol{f}^*(\boldsymbol{x})$ shown in Section 4. We here assume that the integral appearing in the definition of the observation model (5) is well-defined, and defer discussion on conditions for its well-definedness to Appendix C. Let $\boldsymbol{f}(\boldsymbol{x}) \sim \mathcal{GP}(\boldsymbol{m}(\boldsymbol{x}), \mathbf{K}(\boldsymbol{x}, \boldsymbol{x}'))$ be a multivariate GP defined on $\mathcal{X} \subset \mathbb{R}^2$ taking values in $\mathbb{R}^S$. For an arbitrary $k, k' \in \mathbb{N}$ and an arbitrary set of $(k + k')$ points $\boldsymbol{x}_1, \ldots, \boldsymbol{x}_k, \boldsymbol{x}'_1, \ldots, \boldsymbol{x}'_{k'} \in \mathcal{X}$, let

$$\hat{\boldsymbol{f}} = \begin{pmatrix} \bar{\boldsymbol{f}} \\ \bar{\boldsymbol{f}}' \end{pmatrix}, \quad \bar{\boldsymbol{f}} = \begin{pmatrix} \boldsymbol{f}(\boldsymbol{x}_1) \\ \vdots \\ \boldsymbol{f}(\boldsymbol{x}_k) \end{pmatrix}, \quad \bar{\boldsymbol{f}}' = \begin{pmatrix} \boldsymbol{f}(\boldsymbol{x}'_1) \\ \vdots \\ \boldsymbol{f}(\boldsymbol{x}'_{k'}) \end{pmatrix}. \tag{22}$$

By the definition of GP, $\hat{\boldsymbol{f}}$ is a $(k + k')S$-dimensional Gaussian vector. Let

$$\hat{\boldsymbol{m}} = \begin{pmatrix} \bar{\boldsymbol{m}} \\ \bar{\boldsymbol{m}}' \end{pmatrix}, \quad \bar{\boldsymbol{m}} = \begin{pmatrix} \boldsymbol{m}(\boldsymbol{x}_1) \\ \vdots \\ \boldsymbol{m}(\boldsymbol{x}_k) \end{pmatrix}, \quad \bar{\boldsymbol{m}}' = \begin{pmatrix} \boldsymbol{m}(\boldsymbol{x}'_1) \\ \vdots \\ \boldsymbol{m}(\boldsymbol{x}'_{k'}) \end{pmatrix} \tag{23}$$

and

$$\hat{\mathbf{K}} = \begin{pmatrix} \bar{\mathbf{K}} & \bar{\mathbf{K}}'^\top \\ \bar{\mathbf{K}}' & \bar{\mathbf{K}}'' \end{pmatrix}, \quad (\bar{\mathbf{K}})_{ij} = \mathbf{K}(\boldsymbol{x}_i, \boldsymbol{x}_j), \quad (\bar{\mathbf{K}}')_{ij} = \mathbf{K}(\boldsymbol{x}'_i, \boldsymbol{x}_j), \quad (\bar{\mathbf{K}}'')_{ij} = \mathbf{K}(\boldsymbol{x}'_i, \boldsymbol{x}'_j) \tag{24}$$

be the mean vector and the covariance matrix of $\hat{\boldsymbol{f}}$. In the following, we specifically assume that $\boldsymbol{x}'_1, \ldots, \boldsymbol{x}'_{k'}$ are taken to be grid points of a regular grid covering $\mathcal{X}$ and with the grid cell volume

$\Delta$, and consider Riemann sums to approximate those integrals on $\mathcal{X}$ appearing in the formulation of SAGP. We then take the limit $\Delta \to 0$ to derive the posterior GP given areal observations on $\boldsymbol{f}(\boldsymbol{x}) \sim \mathcal{GP}(\boldsymbol{m}(\boldsymbol{x}), \mathbf{K}(\boldsymbol{x}, \boldsymbol{x}'))$.

Consider the observation process yielding observations $\boldsymbol{y}$, defined by

$$\boldsymbol{y} = \bar{\mathbf{A}} \bar{\boldsymbol{f}}' + \boldsymbol{w}, \tag{25}$$

where

$$\bar{\mathbf{A}} = (\ \mathbf{A}(\boldsymbol{x}'_1) \quad \cdots \quad \mathbf{A}(\boldsymbol{x}'_{k'})\ )\Delta, \tag{26}$$

and where $\boldsymbol{w}$ is an $S$-dimensional Gaussian noise vector with mean zero and covariance $\boldsymbol{\Sigma}$. One has

$$\bar{\boldsymbol{\mu}} = \mathbb{E}(\boldsymbol{y}) = \bar{\mathbf{A}} \bar{\boldsymbol{m}}' \tag{27}$$

and

$$\bar{\mathbf{C}} = \mathrm{Cov}(\boldsymbol{y}) = \bar{\mathbf{A}} \bar{\mathbf{K}}'' \bar{\mathbf{A}}^\top + \boldsymbol{\Sigma}, \tag{28}$$

respectively. The posterior of $\bar{\boldsymbol{f}}$ given $\boldsymbol{y}$ is known to be a multivariate Gaussian with mean

$$\bar{\boldsymbol{m}}^* = \bar{\boldsymbol{m}} + \bar{\mathbf{H}}^\top \mathbf{C}^{-1}(\boldsymbol{y} - \boldsymbol{\mu}) \tag{29}$$

and covariance

$$\bar{\mathbf{K}}^* = \bar{\mathbf{K}} - \bar{\mathbf{H}}^\top \mathbf{C}^{-1} \bar{\mathbf{H}}, \tag{30}$$

respectively, where $\bar{\mathbf{H}} = \bar{\mathbf{A}} \bar{\mathbf{K}}'$.

By regarding sums over the $k'$ terms as Riemann sums approximating the corresponding integrals over $\mathcal{X}$, in the limit $\Delta \to 0$, one can replace those sums over $k'$ terms with the corresponding integrals over $\mathcal{X}$. Specifically, one has

$$\begin{aligned}
\boldsymbol{y} &= \sum_{i=1}^{k'} \mathbf{A}(\boldsymbol{x}'_i) \boldsymbol{f}(\boldsymbol{x}'_i) \Delta + \boldsymbol{w} \\
&\to \int_{\mathcal{X}} \mathbf{A}(\boldsymbol{x}) \boldsymbol{f}(\boldsymbol{x})\, d\boldsymbol{x} + \boldsymbol{w},
\end{aligned} \tag{31}$$

$$\begin{aligned}
\bar{\boldsymbol{\mu}} &= \sum_{i=1}^{k'} \mathbf{A}(\boldsymbol{x}'_i) \boldsymbol{m}(\boldsymbol{x}'_i) \Delta \\
&\to \int_{\mathcal{X}} \mathbf{A}(\boldsymbol{x}) \boldsymbol{m}(\boldsymbol{x})\, d\boldsymbol{x} = \boldsymbol{\mu},
\end{aligned} \tag{32}$$

$$\begin{aligned}
\bar{\mathbf{C}} &= \sum_{i,j=1}^{k'} \mathbf{A}(\boldsymbol{x}'_i) \mathbf{K}(\boldsymbol{x}'_i, \boldsymbol{x}'_j) \mathbf{A}(\boldsymbol{x}'_j)^\top \Delta^2 + \boldsymbol{\Sigma} \\
&\to \iint_{\mathcal{X} \times \mathcal{X}} \mathbf{A}(\boldsymbol{x}) \mathbf{K}(\boldsymbol{x}, \boldsymbol{x}') \mathbf{A}(\boldsymbol{x}')^\top\, d\boldsymbol{x}\, d\boldsymbol{x}' + \boldsymbol{\Sigma} = \mathbf{C},
\end{aligned} \tag{33}$$

showing that the mean vector $\bar{\boldsymbol{\mu}}$ and the covariance matrix $\bar{\mathbf{C}}$ of $\boldsymbol{y}$ are reduced in this limit to the vector $\boldsymbol{\mu}$ and the matrix $\mathbf{C}$ defined in (9) and (10), respectively. One also has

$$\begin{aligned}
\bar{\mathbf{H}} &= \Big(\ \textstyle\sum_{i=1}^{k'} \mathbf{A}(\boldsymbol{x}'_i) \mathbf{K}(\boldsymbol{x}'_i, \boldsymbol{x}_1)\Delta \quad \cdots \quad \sum_{i=1}^{k'} \mathbf{A}(\boldsymbol{x}'_i) \mathbf{K}(\boldsymbol{x}'_i, \boldsymbol{x}_k)\Delta\ \Big) \\
&\to \big(\ \textstyle\int_{\mathcal{X}} \mathbf{A}(\boldsymbol{x}') \mathbf{K}(\boldsymbol{x}', \boldsymbol{x}_1)\, d\boldsymbol{x}' \quad \cdots \quad \int_{\mathcal{X}} \mathbf{A}(\boldsymbol{x}') \mathbf{K}(\boldsymbol{x}', \boldsymbol{x}_k)\, d\boldsymbol{x}'\ \big) \\
&= (\ \mathbf{H}(\boldsymbol{x}_1) \quad \cdots \quad \mathbf{H}(\boldsymbol{x}_k)\ ),
\end{aligned} \tag{34}$$

where $\mathbf{H}(\boldsymbol{x})$ is defined in (14). The above calculation shows that in the limit $\Delta \to 0$ the posterior process $\boldsymbol{f}^*(\boldsymbol{x})$ is a multivariate GP with mean function $\boldsymbol{m}^*(\boldsymbol{x})$ and covariance function $\mathbf{K}^*(\boldsymbol{x}, \boldsymbol{x}')$ given by (15) and (16), respectively.

## C  On integrability

In this appendix, we discuss conditions for the observation model (5) to be well defined. Assume that $\mathcal{X} \subset \mathbb{R}^2$ is a bounded Jordan-measurable set, and that elements $a_{s,n}(\boldsymbol{x})$ of $\mathbf{A}(\boldsymbol{x})$ are Riemann integrable on $\mathcal{X}$. (The latter assumption is satisfied when $a_{s,n}(\boldsymbol{x})$ is defined as in (7) with

Jordan-measurable regions $\mathcal{R}_{s,n}$.) Since it is known that a continuous function on $\mathcal{X}$ is Riemann integrable on $\mathcal{X}$, and that a product of Riemann integrable functions is again Riemann integrable, a sufficient condition for the observation model (5) to be well defined is that the prior process $\boldsymbol{f}(\boldsymbol{x}) \sim \mathcal{GP}(\boldsymbol{m}(\boldsymbol{x}), \mathbf{K}(\boldsymbol{x}, \boldsymbol{x}'))$ is sample-path continuous.

The assumption made in Section 3 of the integrability of $\nu_l(\boldsymbol{x})$, $l = 1, \ldots, L$, assures integrability of the mean function $\boldsymbol{m}(\boldsymbol{x}) = \mathbf{W}\boldsymbol{\nu}(\boldsymbol{x})$, which allows us to reduce integrability of the prior process $\boldsymbol{f}(\boldsymbol{x}) \sim \mathcal{GP}(\boldsymbol{m}(\boldsymbol{x}), \mathbf{K}(\boldsymbol{x}, \boldsymbol{x}'))$ to sample-path continuity of the zero-mean process $\boldsymbol{f}(\boldsymbol{x}) \sim \mathcal{GP}(\boldsymbol{0}, \mathbf{K}(\boldsymbol{x}, \boldsymbol{x}'))$ on $\mathcal{X}$. A sufficient condition [1, Theorem 1.4.1] for the sample-path continuity of the zero-mean Gaussian process is that for some $0 < C < \infty$ and $\alpha, \eta > 0$

$$k_{s,s}(\boldsymbol{x}, \boldsymbol{x}) + k_{s,s}(\boldsymbol{x}', \boldsymbol{x}') - 2k_{s,s}(\boldsymbol{x}, \boldsymbol{x}') \leq \frac{C}{|\log \|\boldsymbol{x} - \boldsymbol{x}'\||^{1+\alpha}} \tag{35}$$

holds for all $s \in \{1, \ldots, S\}$ and for all $\boldsymbol{x}, \boldsymbol{x}'$ with $\|\boldsymbol{x} - \boldsymbol{x}'\| < \eta$. If one uses the squared-exponential kernels for $\{\gamma_l\}$, then one can confirm that the above condition is satisfied, and consequently the observation model (5) is well defined.

It should be noted that the sample-path continuity discussed above is different from the mean-square (MS) continuity. A process $\boldsymbol{f}(\boldsymbol{x})$ is said to be MS continuous at $\boldsymbol{x} = \boldsymbol{x}_*$ if for any sequence $\boldsymbol{x}_k$ converging to $\boldsymbol{x}_*$ as $k \to \infty$ it holds that $\mathbb{E}[\|\boldsymbol{f}(\boldsymbol{x}_k) - \boldsymbol{f}(\boldsymbol{x}_*)\|^2] \to 0$ as $k \to \infty$. A necessary and sufficient condition for a random field to be MS continuous at $\boldsymbol{x}_*$ is that its covariance function $\mathbf{K}(\boldsymbol{x}, \boldsymbol{x}')$ is continuous at the point $\boldsymbol{x} = \boldsymbol{x}' = \boldsymbol{x}_*$ [2, Appendix 10A], which in the case of Gaussian processes is weaker than the above sufficient condition for the sample-path continuity.

## D   Description of real-world areal data sets

We used the real-world areal data sets from NYC Open Data [4] and Chicago Data Portal [5] to evaluate the proposed model. These data sets are collected and released for improving city environments, and consist of a variety of categories including social indicators, land use, and air quality. Details of the areal data sets we used in the experiments are listed in Table 3. The number of data sets in New York City and Chicago are 10 and 3, respectively. Each data set is associated with one of the predefined geographical partitions. The number of partition types in New York City and Chicago are 4 and 2, respectively. Table 3 shows the respective partition names and the number of regions in the corresponding partition. These data sets are gathered once a year at the time ranges shown in Table 3; the values of data were divided by the number of observation times. Then, the data were normalized so that each variable in each city has zero mean and unit variance.

## E   Results

Table 4 shows MAPE and standard errors for GPR, 2-stage GP, SLFM, and SAGP, where the experiments for Crime data set in Chicago have not been conducted because the coarser version for training is not available online. For all data sets, SAGP achieved the comparable or better performance than the other methods.

## Footnotes

[4]https://opendata.cityofnewyork.us

[5]https://data.cityofchicago.org/

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

Table 3: Real-world areal data sets.

(a) New York City

| Data | Partition | #regions | Time range |
|------|-----------|----------|------------|
| PM2.5 | UHF42 | 42 | 2009 – 2010 |
| Poverty rate | Community district | 59 | 2009 – 2013 |
| Unemployment rate | Community district | 59 | 2009 – 2013 |
| Mean commute | Community district | 59 | 2009 – 2013 |
| Population | Community district | 59 | 2009 – 2013 |
| Recycle diversion rate | Community district | 59 | 2009 – 2013 |
| Crime | Police precinct | 77 | 2010 – 2016 |
| Fire incident | Zip code | 186 | 2010 – 2016 |
| 311 call | Zip code | 186 | 2010 – 2016 |
| Public telephone | Zip code | 186 | 2016 |

(b) Chicago

| Data | Partition | #regions | Time range |
|------|-----------|----------|------------|
| Crime | Police Precinct | 25 | 2012 |
| Poverty rate | Community district | 77 | 2008 – 2012 |
| Unemployment rate | Community district | 77 | 2008 – 2012 |

Table 4: MAPE and standard errors for the prediction of fine-grained areal data in New York City and Chicago. The numbers in parentheses denote the number $L$ of the latent GPs estimated by the validation procedure. The single star ($\star$) and the double star ($\star\star$) indicate significant difference between SAGP and other models at the levels of $P$ values of $< 0.05$ and $< 0.01$, respectively.

(a) New York City

| | GPR | 2-stage GP | SLFM | SAGP |
|------|-----|-----------|------|------|
| PM2.5 | $0.072 \pm 0.010$ (–) | $0.042 \pm 0.005$ (–) | $0.036 \pm 0.005$ (6) | $\mathbf{0.030 \pm 0.005}^{\star}$ (5) |
| Poverty rate | $0.344 \pm 0.046$ (–) | $0.210 \pm 0.022$ (–) | $0.207 \pm 0.025$ (4) | $\mathbf{0.177 \pm 0.019}^{\star\star}$ (3) |
| Unemployment rate | $0.319 \pm 0.036$ (–) | $0.193 \pm 0.021$ (–) | $0.195 \pm 0.024$ (3) | $\mathbf{0.165 \pm 0.020}^{\star}$ (3) |
| Mean commute | $0.131 \pm 0.020$ (–) | $0.068 \pm 0.009$ (–) | $0.057 \pm 0.007$ (4) | $\mathbf{0.050 \pm 0.007}$ (6) |
| Population | $0.577 \pm 0.104$ (–) | $0.389 \pm 0.033$ (–) | $0.337 \pm 0.039$ (3) | $\mathbf{0.295 \pm 0.033}^{\star}$ (3) |
| Recycle diversion rate | $0.353 \pm 0.049$ (–) | $0.236 \pm 0.034$ (–) | $0.222 \pm 0.032$ (4) | $\mathbf{0.211 \pm 0.029}$ (4) |
| Crime | $0.860 \pm 0.102$ (–) | $0.454 \pm 0.075$ (–) | $0.401 \pm 0.053$ (2) | $\mathbf{0.379 \pm 0.055}^{\star\star}$ (3) |
| Fire incident | $1.097 \pm 0.097$ (–) | $0.746 \pm 0.084$ (–) | $0.500 \pm 0.052$ (4) | $\mathbf{0.396 \pm 0.038}^{\star\star}$ (3) |
| 311 call | $0.083 \pm 0.004$ (–) | $0.070 \pm 0.004$ (–) | $0.061 \pm 0.004$ (6) | $\mathbf{0.052 \pm 0.003}^{\star\star}$ (3) |
| Public telephone | $0.131 \pm 0.008$ (–) | $0.083 \pm 0.008$ (–) | $0.086 \pm 0.008$ (4) | $\mathbf{0.080 \pm 0.007}$ (6) |

(b) Chicago

| | GPR | 2-stage GP | SLFM | SAGP |
|------|-----|-----------|------|------|
| Poverty rate | $0.599 \pm 0.099$ (–) | $0.380 \pm 0.060$ (–) | $0.335 \pm 0.052$ (2) | $\mathbf{0.278 \pm 0.032}^{\star\star}$ (2) |
| Unemployment rate | $0.478 \pm 0.047$ (–) | $0.318 \pm 0.032$ (–) | $0.278 \pm 0.025$ (2 ) | $\mathbf{0.231 \pm 0.021}^{\star}$ (2) |