[Reviews · NeurIPS 2019]

Reviewer 1



The paper proposes to use integral/summed observations within the Gaussian process framework. For regression, the outcome of this model is neat because all operators are linear. The paper is clear in its presentation, except for a few points listed below. 1, In terms of originality, the paper is a useful application of block kriging and Murray-Smith [35]. However, the paper currently lack references to and discussions on these two works. In additional, eq (4) is an instance of the linear mode of coregionalisation, and references to this is missing. 2, The paper uses numerical approximation of integrals for integral observations (line 198). This is a practical and common approach. However, it'll be contribute to the paper significantly if the authors can propose an alternative, given that this paper is primarily about integral observations. 3. In the experiment, the observations are rates, but the paper uses a regression model. A warp GP model or a more appropriate likelihood would be more appropriate. 4. The experiment compares SAGP with GPR, 2-stage GP and SLFM. I am not clear the relevance of comparing with SLFM. It'll be better if the paper can have a more thorough comparison with 2-stage GP instead, for example, by examining the variances of the predictions and also the significance of the learned hyper-parameters and perhaps also the computational time complexity of the two methods. Clarity points 1. Line 125: Is the domain "a set of cities" rather than just "a city"? 2. Line 302: It is not clear how the transfer learning across multiple cities is achieved. Are these just treated as different locations? In this case, the correlation between data from the two cities would be very minute, unless there is a significant long-scale component in the (stationary) covariance function. [35] Roderick Murray-Smith, Barak A. Pearlmutter: Transformations of Gaussian Process Priors. Deterministic and Statistical Methods in Machine Learning 2004: 110-123

Reviewer 2



Originality: In order to deal with multivariate data the paper proposes to use a particular case of the Linear Model of Coregionalization (LMC [1]). For dealing with spatially aggregated data the paper proposes to use a Gaussian likelihood around the weighted aggregated value of the prediction across regions, resulting in a intractable marginal likelihood which is approximated by using Riemann sums. These are very straight-forward ideas, nevertheless they are effective. Quality: The paper is technically correct and the experimental section is well done. Clarity: The paper is well written and easy to follow for the most part. Only the section concerning the extension to multiple domains needs more details in my opinion. For instance, is normalization needed across domains? Significance: It is a good paper and the experimental section shows the importance of choosing the right likelihood function for the data, but I find the contribution rather limited in scope. [1] Álvarez, M. A., Rosasco, L. & Lawrence, N. D. Kernels for Vector-valued Functions: A Review (Now Publishers Incorporated, 2012). [2] Ho Chung Law, Dino Sejdinovic, Ewan Cameron, Tim Lucas, Seth Flaxman, Katherine Battle, and Kenji Fukumizu, Variational learning on aggregate outputs with gaussian processes, Advances in Neural Information Processing Systems 31 (NeurIPS 2018).

Reviewer 3



# [Updated after author feedback] I thank the authors for their feedback. My suggestions for improvements were only sparingly addressed, but I will keep my score as it is. My request for updating table 1 was perhaps unclear. I appreciate that you focus on just three features as they are important in socioeconomics, but I would like to see the same results for all datasets for both cities (thus ten columns for New York, three columns for Chicago). Only choosing a subset, even though it can be motivated from an application point-of-view, seems arbitrary and makes one suspect cherrypicking. This is, hopefully, an unfair suspicion, so why not include a large table with results from all features? Even if a feature is not interesting from an application point-of-view, it is still important for judging the performance of the model. # Summary The paper is concerned with modelling a multivariate function from multiple areal datasets at different granularities. The authors propose a model, based on Gaussian processes (GPs), that handles data defined as regions of the input space. The model initially follows the standard multivariate GP strategy by defining independent latent GPs, which are then linearly combined to form a multivariate dependent GP. To handle data at different granularities, observations are assumed to be area integrals of the multivariate GP. This allows the model to infer function values on a fine-scale from coarsely sampled data. The model also naturally handles data from different domains by sharing the latent GPs across the domains. The proposed model is evaluated using a total of 13 datasets from two cities, each with varying granularity. A refinement task, estimating small-scale structure from large-scale, is considered in two different set-ups: refining data within a single city and refining data across cities by utilising the transfer learning capabilities of the model. The model shows performance improvements over both baselines and competing models. # Quality The paper appears technically sound. The sections deriving the model and inference are detailed, yet concise, and further information is provided in the supplementary. Related work is adequately cited, and the shortcomings of these methods are nicely outlined. The main difference between the proposed model and the semi-parametric latent factor model (SLFM) it builds upon is clearly described. I think a reference to Alvarez, Rosasco, & Lawrence, "Kernels for vector-valued functions: A review", Foundations and Trends in Machine Learning (2012) would, however, be appropriate to include. The experiments are interesting and convincing. Is there a reason why only some domains have been included in table 1 and 2? I would like to see similar results for all domains in the two cities and for both tasks. Right now, one can get the feeling that the results are cherry-picked. I think it would also be interesting to see if the performance gain saturates as more cities or domains are added, though this is probably better left for future work. # Clarity The paper reads quite well. The structure is clear and good and I found it surprisingly easy to follow the definition of the model despite the dense notation. Good job! Figure 2 very nicely explains the model. I found it very helpful. # Originality To my knowledge, the introduction of spatial aggregation of the input space is indeed novel, though perhaps not overly so. The idea might seem simple, but it is good and the resulting model elegant. # Significance The proposed model seamlessly shares information between regions of different granularities as well as between cities, which makes it both interesting and important, particularly for the geostatistical community. While focusing on a somewhat specific problem, it is a solid paper that I think would be welcomed by the NeurIPS community.

[Author Response · NeurIPS 2019]

**Response to reviewers' comments (Paper ID: 1711)**

We would like to thank the reviewers for their feedback and constructive comments, which we respond to below. The blue parts are the value-added contributions of our paper. We would be grateful if you could check it carefully.

**To Reviewer 1.**

1. **Related work:** The concept related to our aggregation processes has been used in several methods [5, 35]. The linear model of coregionalization (LMC) is the general formulation for multivariate modeling in geostatistics; our baseline (i.e., SLFM) is an instance of LMC, in which latent functions are assumed to be Gaussian processes (as described in [36]). As the reviewer commented, our proposal is based on these two key concepts. We still believe that our paper makes significant technical contributions: 1) integrating aggregation processes with multivariate Gaussian processes that is defined by LMC-based approach; 2) parameter estimation based on the marginal likelihood in which latent GPs $g(x)$ and $f(x)$ are analytically integrated out; 3) explicit derivation of the posterior GP given multivariate areal observations (see also Appendix B). We will cite the references and discuss them.

2. **Integral over regions:** As the reviewer suggested, it could be an additional contribution to consider an alternative to performing integral over regions. However, we strongly believe that our paper contains enough valuable contributions (as described above); moreover our experiments on real-world data sets from multiple cities show that our model significantly improves the performance in predicting the fine-grained mappings from coarse-grained areal data. This is very helpful for many disciplines including socio-economics [24], epidemiology [25], public security [2], public health [13], and urban planning [34].

3. **Handling observations:** We agree with the reviewer in that introduction of nonlinear link functions (as in the warped GP) and/or alternative likelihoods might help handle some kinds of observations (e.g., rates). One can find, however, in the literature many successful models (e.g., [26]) based on similar assumptions to ours.

4. **Compared methods:** As shown in Table 1 of the manuscript, we confirmed that our model (i.e., SAGP) yields better prediction performance than all the baselines, that is, GPR, 2-stage GP, and SLFM. The detailed comparison with SLFM is important and reasonable, because SAGP is regarded as the extension of SLFM; thus this can clarify the contribution of SAGP. Meanwhile, we also agree with the reviewer in that a more thorough examination from another point of view (e.g., prediction variances) might be helpful, which is one of our future works.

5. **Definition of domain:** In the "Areal data" paragraph (lines 121–130), we describe the case of a single domain (i.e., a city); thus $\mathcal{X}$ denotes a city. For the case of multiple domains, a set of cities is denoted by $\mathcal{X}^{\mathrm{u}}$ on line 176.

6. **Transfer learning across cities:** Given the latent GPs $\{g_l(x)\}_{l=1}^L$, the observations $y^{(1)}$ and $y^{(2)}$ from respective cities are treated independently (see Figure 2(b)). Although $y^{(1)}$ and $y^{(2)}$ are not directly correlated across cities, the *shared* covariance functions $\{\gamma_l(x, x')\}_{l=1}^L$ for the latent GPs can be learnt by transfer learning based on the data sets from multiple cities; thus the spatial correlation for each data set could be more appropriately output via the covariance functions, even if we have only a few data sets available for a single city. We will clarify it.

**To Reviewer 2.**

1. **Difference from LMC [36]:** Please see the 1st response to **Reviewer 1**.

2. **Extension to multiple domains:** In the experiments, the data were normalized so that each variable in each city has zero mean and unit variance. We will clarify it in our proceeding. Please see also the 6th response to **Reviewer 1** for additional information about the transfer learning across multiple domains.

3. **Significance:** As described in the 2nd response to **Reviewer 1**, we believe that our paper provides significant contributions not only to the field of machine learning but also to various other fields. Moreover, our formulation provides a general framework for handling aggregated data and offers a potential research direction that can be explored in future work; for instance, it has the capability to consider the data aggregated on a higher dimensional input domain $\mathcal{X}$, e.g., spatio-temporal aggregated data.

4. **Using inducing points:** Extension of inference algorithm with inducing points is possible, which is one of the options for the efficient computation. However, the computation complexity $O(|\mathcal{P}_s||\mathcal{P}_{s'}||\mathcal{D}|)$ of our model (see line 230) is not too high; actually the average computation times for inference were 1728.2 and 115.1 seconds for the data sets from NYC and CHI, respectively, where the experiments were conducted on a 3.1 GHz Intel Core i7. The results show that our inference algorithm is efficient enough to run in realistic-time scales.

**To Reviewer 3.**

1. **Related work:** We will cite the reference [36]. Please see the 1st response to **Reviewer 1** for more details.

2. **Experiments:** In this paper, we focused on the refinement task of poverty, crime, and PM2.5 data sets. The main reasons are as follows: 1) refining these data sets is strongly desired in the practice of socio-economics [2, 24] and public health [13]; 2) our experiments basically replicate the experimental setup in the existing work (2-stage GP) that has proposed in [26]. We agree with the reviewer in that the extensive experiments on more data sets from more cities are left for future work.

**References**

[35] R. Murray-Smith and B. A. Pearlmutter. Transformations of Gaussian process priors. In *DSMML*, pages 110–123, 2004.

[36] M. A. Álvarez, L. Rosasco, and N. D. Lawrence. Kernels for vector-valued functions: A review. *Foundations and Trends ® in Machine Learning*, 4(3):195–266, 2012.


[Meta-Review · NeurIPS 2019]

All reviewers agreed that the paper presents an original idea for dealing with spatially aggregated data by constructing a suitable likelihood used together with a GP framework. Experimentally the method seems to be effective. For the final version please address all reviewers' comments.